# The Potential of Ectomycorrhizal Fungi to Modulate below and Aboveground Communities May Be Mediated by 1-Octen-3-ol

**DOI:** 10.3390/jof9020180

**Published:** 2023-01-29

**Authors:** Inês Ferreira, Teresa Dias, Cristina Cruz

**Affiliations:** cE3c—Centre for Ecology, Evolution and Environmental Changes & CHANGE—Global Change and Sustainability Institute, Faculdade de Ciências, Universidade de Lisboa, Campo Grande, Bloco C2, 1749-016 Lisboa, Portugal

**Keywords:** 1-octen-3-ol, C-8 volatiles, fungal volatiles, ectomycorrhizal fungi, *Terfezia*, Cistaceae

## Abstract

It is known that ectomycorrhizal (ECM) fungi can modulate below and aboveground communities. They are a key part of belowground communication as they produce a vast array of metabolites, including volatile organic compounds (VOCs) such as 1-octen-3-ol. Here, we tested if the VOC 1-octen-3-ol may be involved in the ECM fungal mechanisms that modulate below and aboveground communities. For that, we conducted three in vitro assays with ECM fungi and the 1-octen-3-ol volatile to (i) explore the effects of mycelium growth of three ECM species, (ii) investigate the impact on the germination of six host Cistaceae species, and (iii) study the impact on host plant traits. The effects of 1-octen-3-ol on mycelium growth of the three ECM species depended on the dose and species: *Boletus reticulatus* was the most sensitive species to the low (VOC) dose, while *T. leptoderma* was the most tolerant. In general, the presence of the ECM fungi resulted in higher seed germination, while 1-octen-3-ol resulted in lower seed germination. The combined application of the ECM fungus and the volatile further inhibited seed germination, possibly due to the accumulation of 1-octen-3-ol above the plant species’ threshold. Seed germination and plant development of Cistaceae species were influenced by ECM fungal volatiles, suggesting that 1-octen-3-ol may mediate changes in below and aboveground communities.

## 1. Introduction

Although it remains unclear when and how ectomycorrhizal (ECM) fungi mediate the direction and strength of feedback in plant communities, it is consensual that they can modulate below and aboveground communities [1]. Several factors can contribute to the explanation of the effects of ECM fungi on communities, namely the mediation of plant–soil feedbacks [2] and of plant defenses [3,4], plant facilitation [5] and communication with other soil microorganisms [6,7].

Volatile organic compounds (VOCs) are a vital communication strategy among individuals of a species and of different kingdoms [8]. Among the most common fungal VOCs, there are the eight carbon compounds (C-8), which result from the oxidation of linoleic acid [9]. These compounds are also known as oxylipins, and play key roles in regulating fungal morphogenesis, plant interactions and biocontrol [10]. Among these, 1-octen-3-ol, also known as the “mushroom alcohol” due to its characteristic mushroom flavor [9,11], is one of the most abundant VOCs produced by fungi [3] and is considered a signaling molecule, produced in different fungal structures, conidial masses, sporocarps and hyphae [12,13].

The C-8 volatile oxylipin 1-octen-3-ol has an important role in fungal interactions, presenting a regulatory effect on the physiology and development of several fungi from different genera [12]. This compound can inhibit fungal growth and spore germination of *Botrytis cinerea* [3], *Penicillium paneum* [13], *Penicillium chrysogenum*, *Monilinia fructicola* [14], *Fusarium tricinctum* and *Fusarium oxysporum* [11]. Furthermore, 1-octen-3-ol also inhibits the growth of several bacterial species (e.g., *Staphylococcus aureus*, *Bacillus subtilis*, *Staphylococcus epidermidis*, *Escherichia coli*, *Pseudomonas aeruginosa*) [11], while, in saprophytic species such as *Agaricus bisporus*, this volatile is involved in bacterial stimuli for primordium formation [15].

Besides being involved in fungi–fungi and fungi–bacteria interactions, 1-octen-3-ol is also involved in fungi–plant interactions. For example, 1-octen-3-ol inhibits the fungal pathogen *Botrytis cinerea* and contributes to enhancing plant resistance by the induction of defense signaling cascades [9,10]. For instance, treating *Zea mays* seeds with 100 µL of 1-octen-3-ol in the form of the patented product (Omega™) resulted in season-long improvements in shoot and root growth, similar to those obtained when applying *Trichoderma* spp. [16]. Low concentration (0.05 µL/L) of this VOC also showed plant growth-promoting activity in tomato plants [17]. Therefore, 1-octen-3-ol can stimulate or inhibit the growth of neighboring organisms (other fungi, bacteria and plants) in a dose-dependent manner [18].

1-octen-3-ol is considered a biomarker characteristic for symbiotic fungi, being emitted in high doses by these fungi, while a low emission of this compound can be observed in non-symbiotic fungi [19]. In agreement, and although 1-octen-3-ol was first identified in ectomycorrhizal (ECM) fungi in their fruitbodies [20], mycelium [21] and during ectomycorrhiza synthesis [22], its effects on spore germination, mycelium growth and primordia initiation of these fungi have not yet been documented. Furthermore, the study of interactions between ECM fungal VOCs, such as 1-octen-3-ol, and their host plants have been insufficiently studied.

ECM fungi play an important role in the ecology of Mediterranean shrublands by helping plants to survive in such nutrient-poor soils that are common in these ecosystems [23]. In this biome, ECM fungi have been found to be associated with many plant species such as *Cistus*, *Quercus* and *Pinus* [24,25,26]. *Cistus* sp. are frequent in Mediterranean Basin plant communities and are well adapted to the harsh conditions of Mediterranean shrublands [27]. In addition, *Cistus* sp. have been found to form mycorrhizal relationships with various ECM fungal species, including those in the genus *Terfezia*, *Lactarius* and *Boletus* [28,29,30,31,32]. Therefore, the present study aims to evaluate the effects of the fungal volatile 1-octen-3-ol in ECM fungi and their host plants in different stages of their interactions towards symbiosis establishment. We hypothesized that 1-octen-3-ol has the potential to modulate below and aboveground communities due to different thresholds of ECM fungal and plant species. For that, we performed three in vitro assays to (i) explore the effects of 1-octen-3-ol on mycelium growth of three ECM species: *Terfezia leptoderma*, *Lactarius deliciosus* and *Boletus reticulatus*; (ii) investigate the impact of 1-octen-3-ol and the two ECM species on the germination of six host Cistaceae species: *Cistus albidus* L., *C. ladanifer* L., *C. salviifolius* L., *C. psilosepalus* Sweet, *Halimium halimifolium* (L.) Willk. and *Tuberaria guttata* (L.) Fourr.); and (iii) study the impact of 1-octen-3-ol or ECM fungus (*Terfezia arenaria*) on host plant *Cistus salviifolius* traits for nine months. 

## 2. Materials and Methods

### 2.1. Fungal Material

Mature *Terfezia arenaria*, *T. leptoderma*, *Lactarius deliciosus* and *Boletus reticulatus* sporocarps were harvested from different locations in Portugal (Alentejo and Leiria regions) during autumn and spring seasons of 2019. Specimens were freed from substrate debris at the site and further cleaned in the laboratory. The sporocarps were identified by their morphological macro and microscopic characteristics and by following several authors [33,34] and online keys (http://www.mycokey.com/ accessed on 13 March 2019) [35]. In the case of the *Terfezia* species, due to the similarity between species, the samples were further identified by molecular analysis.

The mycelia of three ectomycorrhizal species (*T. leptoderma*, *L. deliciosus* and *B. reticulatus)* were isolated from sporocarps on potato dextrose agar (PDA) (VWR), pH 5.5. Cultures were incubated at 25 °C, transferred onto fresh medium every 3 months and maintained at 4 °C. Thereafter, the sporocarp samples were dried at 35 °C in a forced ventilation oven (Lab Companion, Model OF-11E) until constant mass. Dried fungal materials were powdered in a porcelain mortar and kept in brand-new sealed polyethylene bags under dry conditions. For sporal inoculum preparation, previously powdered, 2.4 g of dried *T. arenaria* ascocarps was added to 400 mL of sterile distilled water and left to shake overnight at 23 °C ± 1 °C in the dark before use. A 100-µL aliquot was taken to count the spore’s concentration under the microscope, which was found to be 1 × 10^8^ of spores mL^−1^.

### 2.2. Plant Material

Seeds from *Cistus salviifolius* L., *C. psilosepalus* Sweet and *Tuberaria guttata* (L.) Fourr. plants growing in the Leiria region (Portugal) were collected in September 2018 in a *Pinus pinaster* forest area with a natural shrub understory. The seeds were dried at room temperature and kept in the dark until use. Seeds of *C. albidus* L., *C. ladanifer* L. and *Halimium halimifolium* (L.) Willk. were purchased from Sementes de Portugal and kept in the dark until use.

To break seed dormancy, seed scarification was performed by rubbing the seeds over a rough surface (i.e., sandpaper) and then heating them at 105 °C in an oven (Lab Companion, Model OF-11E) for 10 min and leaving them to cool down until room temperature. The seeds were hydrated for 10 min in sterile distilled water, surface sterilized by immersion in 30 % H_2_O_2_ for 30 min and washed three times with sterile distilled water followed by another immersion in a 20% bleach solution with 3 drops of Tween for 5–10 min. Afterwards, the seeds were washed three times in sterilized water. Seeds were then used in the germination tests.

### 2.3. VOC Assays

To understand the potential of 1-octen-3-ol to modulate below and aboveground communities, we conducted three in vitro assays to test the effects of 1-octen-3-ol on the development of ECM fungal species and host plants, namely on ECM mycelium growth, Cistaceae sp. seeds and in *C. salviifolius* traits.

In the following assays, three doses of 1-octen-3-ol were tested: 0 µg (designated as CT); 0.17 µg (designated as VOC_low); and 1280 µg (designated as VOC_high). The doses were selected from the range reported in several studies using 1-octen-3-ol [12,36,37,38,39]. The low dose we used took into consideration the positive results obtained in previous works, while the high dose was higher than all the doses reported.

#### 2.3.1. Mycelium Development

To evaluate the effect of the 1-octen-3-ol dose on the development of the mycelium of the three ECM fungi (*Terfezia leptoderma*, *Lactarius deliciosus* and *Boletus reticulatus*), we tested three doses per plate: 0 µg (CT); 0.17 µg (VOC_low); and 1280 µg (VOC_high). For that, we used PDA (VWR) with pH 5.5 as the culture medium (without antibiotics addition). On the border of the Petri dishes (9 cm diameter) with the PDA medium, we added 10 µL of 1-octen-3-ol (VOC) in different doses.

The solution of 1-octen-3-ol at 1 µM was prepared using 1-octen-3-ol 98% (Alfa Aesar CAS 3391-86-4) diluted in Chloroform 99% (Merck CAS 67-66-3) and sterilized by filtration (0.2 μm, 47 mm membrane filter, Minisart^®^ NML, Sartorius, Gottingen, Germany). The mycelial plugs with 0.25 cm^2^ of *T. leptoderma*, *L. deliciosus* and *B. reticulatus* were transferred from 1-month pure cultures (in PDA) and placed in the centre of the Petri dishes of each culture medium. Each treatment was replicated 10 times (N = 10 Petri dishes).

The ECM fungal cultures were placed in a growth chamber at 23 °C ± 1 °C in the dark. Colonies’ growths were measured every 7 days for 63 days by measuring mycelia radial growth at the bottom of the Petri dish.

#### 2.3.2. Effect of VOC and ECM Fungi on Cistaceae Species Germination

To evaluate the effect of 1-octen-3-ol (VOC) on the germination of plant hosts to the ECM fungi, we used six Cistaceae species (*Cistus albidus* L., *C. ladanifer* L., *C. salviifolius* L., *C. psilosepalus* Sweet, *Halimium halimifolium* (L.) Willk. and *Tuberaria guttata* (L.) Fourr.) and two ECM fungal species (*Terfezia leptoderma* and *Lactarius deliciosus*) whose growth was not affected by the low VOC dose. Given that high concentrations of 1-octen-3-ol can damage plants [19], we used the lower doses (i.e., 0 µg; 0.17 µg per plate). Four treatments were tested for each Cistaceae sp., with five replicates (i.e., 9 cm Ø Petri dishes) for each species and treatment (Figure 1).

PDA pH 5.5 culture medium (without antibiotics addition) was used for the germination of Cistaceae seeds with ECM mycelium or the VOC. For all treatments, in each Petri dish, we placed 4 seeds previously sterilized, scarified and heated to break dormancy (as previously explained in Section 2.2). For the VOC treatment, on the center of the Petri dish, we added 10 µL of 1 µM 1-octen-3-ol solution (0.17-µg dose per plate). In the CT, we added only 10 µL of chloroform. For the ECM fungi treatments, mycelium plugs (0.25 cm^2^) of *T. leptoderma* (Tlep) or *L. deliciosus* (Ldel) were placed in the center of the Petri dishes. In the control (CT) Petri dishes, only the seeds were used (i.e., ECM fungi and VOC were absent). Five replicates of each treatment were prepared. The Petri dishes were placed in a growth chamber at 23 °C ± 1 °C in the dark with 60 ± 5% relative humidity, and germination was evaluated after 35 days. 

#### 2.3.3. Effect of VOC and ECM on *Cistus salviifolius* Germination and Development

Although we used *T. leptoderma* in the previous assays, we did not have a viable spore inoculum to carry out this assay. Since both *T. arenaria* and *T. leptoderma* are characteristic of acid soils and are able to form in vitro mycorrhizal associations with *Cistus salviifolius* [28,40], we used *T. arenaria* in this third assay; based on the tolerance of *T. leptoderma* and of *C. salviifolius* to the low VOC dose, we used *Cistus salviifolius* as the plant host and *T. arenaria* as the ECM fungus to evaluate the effect of 1-octen-3-ol (VOC) during symbiosis establishment on host plant traits. The following four treatments were implemented: CT—control treatment only with *Cistus salviifolius* seeds; VOC—treatment with 10 µL of 1-octen-3-ol low dose (0.17 µg per plate); Ta—treatment with 10 mL of *T. arenaria* sporal inoculum (10^8^ mL^−1^); and TaVOC—treatment with 10 µL of 1 µM 1-octen-3-ol solution (0.17 µg dose per plate) and 10 mL of *T. arenaria* sporal inoculum (10^8^ mL^−1^) (Figure 2).

Germination was performed in polypropylene transparent microboxes (90 mm Ø and 120 mm in height) with a cover without filters. Each box contained 100 mL of sterilized substrate (peat:perlite mixture, 3:1 *v/v*) and 50 mL of distilled water was autoclaved at 121 °C for 60 min. In a flow chamber, ten previously disinfected seeds (as described in Section 2.2) were placed in each container.

For the treatments with the ECM fungus (Ta and TaVOC), after the ten seeds were distributed in the microboxes, 10 mL of *Terfezia* sporal inoculum was distributed in each microbox of the two treatments with Terfezia. In the treatments with the VOC (VOC and TaVOC), 10 µL of 1-octen-3-ol 1 µM was pulverized inside each microbox. As control, we used microboxes only with *Cistus salviifolius* seeds (CT), and we added only 10 µL of chloroform. 

The microboxes (N = 18 per treatment) were kept in a growth chamber without direct light, a 16 h dark/8 h light photoperiod and 25 °C/20 °C (±2 °C) day/night temperature. The bottom part of the microboxes, containing the substrate with the seeds and the spores, was covered with aluminum foil to decrease light incidence in this area. The number of germinated plants was counted every month for three months. In the third month, root samples from six microboxes per treatment were collected to confirm mycorrhization by microscopic characterization. At six and nine months, six microboxes per treatment were selected randomly, and the following plant traits were measured: shoot length, fresh shoot weight, root length, fresh root weight, number of branches per shoot and number of leaves per shoot. The root and shoot fresh weight were measured on an analytical scale (Radwag) with 0.0001 g resolution. The shoot and root length were measured with a ruler, and the number of number of branches and number of leaves were counted per shoot.

### 2.4. Statistical Analysis

Statistical analysis was conducted using Microsoft Excel 2019/XLSTAT-Premium (Version 2021.4.1, Addinsoft, Inc., Brooklyn, NY, USA). Since our numerical variables did not follow a normal distribution, the Kruskal–Wallis test was selected. Differences between the treatments were compared using the Kruskal–Wallis one-way analysis of variance. Multiple pairwise comparisons were performed using the Dunn’s test (*p* < 0.05).

## 3. Results

### 3.1. Mycelium Growth

Although no antibiotics were added to the PDA medium, contaminations were not observed on the pure cultures of the ECM fungi. While the high 1-octen-3-ol dose (VOC_high) fully inhibited the mycelium growth of all three ECM species, the low 1-octen-3-ol dose (VOC_low) resulted in different responses in the ECM species (Figure 3). When compared to the control, the mycelium growth of *B. reticulatus* was inhibited (Figure 3a), that of *L. deliciosus* showed a non-significant tendency for inhibition (Figure 3b) and that of *T. leptoderma* was not affected (Figure 3c). Therefore, the mycelium growth of these three ECM species showed a sensitivity gradient to the 1-octen-3-ol low dose (VOC_low), ranging from no effect to growth inhibition. From the tested ECM species, *Boletus reticulatus* was shown to be the most sensitive species to the low VOC dose, while *T. leptoderma* was the most tolerant. 

### 3.2. Cistaceae Species Germination

The germination of the six Cistaceae species (*Cistus albidus*, *C. ladanifer*, *C. psilosepalus*, *C. salviifolius*, *Halimilium halimifolium* and *Tuberaria guttata*) after 35 days in co-culture with *Terfezia leptoderma, Lactarius deliciosus* or VOC (lower 1-octen-3-ol dose) showed distinct responses to the volatile and ECM mycelium (Figure 4, Appendix A). We observed seed germination for all Cistaceae sp. in the four treatments. Germination without the volatile and ECM mycelium (i.e., CT) did not result in higher germination rates for any of the Cistaceae sp., and, in *T. guttata*, it resulted in the lowest germination rate (40%).

The low VOC dose had contrasting effects on the Cistaceae sp. germination. For half of the Cistaceae sp. tested (*C. albidus*, *C. psilosepalus* and *H. halimifolium*), the presence of the low VOC dose resulted in the lowest germination rates, ranging from 20% in *C. albidus* and *C. psilosepalus* to 30% in *H. halimifolium*. In *C. ladanifer* and *T. guttata*, there was a non-significant tendency to stimulate germination, resulting in high germination rates for these species (60% and 85%, respectively). In *C. salviifolius*, germination was similar in all treatments, including the low VOC dose. Altogether, the tested Cistaceae sp. showed different sensitivities to the low 1-octen-3-ol dose.

Finally, the presence of the ECM fungi never had a negative effect on the *Cistaceae* sp. germination; it had no effect (e.g., *C. ladanifer* and *C. salviifolius*) or it stimulated germination (e.g., *C. albidus*, *C. psilosepalus* and *T. guttata* with Ldel and *H. halimifolium* with Tlep).

### 3.3. Cistus salviifolius Traits

Although there were no differences in *C. salviifolius* germination between the low 1-octen-3-ol dose (VOC) and the control (CT) after three months, the combined addition of the volatile with the ECM fungus (TaVOC) inhibited the germination by ~60% (in relation to the CT). By contrast, adding the ECM fungus alone (Ta) stimulated *C. salviifolius* germination by ~35% (in relation to the CT) (Figure 5). 

From the analysis of the root samples in the third month, for the treatments with T. arenaria inoculum, mycorrhizal structures were detected in the initial stage. From the six *C. salviifolius* traits evaluated six and nine months after the beginning of the assay, only the number of lateral shoots and the number of leaves showed differences between the treatments (Table 1). At month six, the combined addition of the volatile with the ECM fungus (TaVOC) stimulated the number of lateral shoots (*p* < 0.05), and no lateral shoots could be observed in the CT plants. Still at month six, the addition of the volatile (VOC) or the ECM fungus (Ta) resulted in lower numbers of lateral shoots, similar to the CT.

Nine months after the beginning of the assay, TaVOC still resulted in the highest number of lateral shoots, and so did Ta, and both were higher than the CT (*p* < 0.05). The number of leaves, which was similar for all treatments at month six, showed differences at month nine: Ta and TaVOC plants had more leaves that the CT plants (*p* < 0.05). In addition, for the shoot and root fresh weight, there was a tendency for the TaVOC plants to show higher values.

## 4. Discussion

The lower dose of 1-octen-3-ol inhibited the mycelium growth of *B. reticulatus* but not of *L. delicious* and *T. leptoderma*. This may reflect distinct responses of ECM species to 1-octen-3-ol, where ECM species may differ in their thresholds to the volatile. Therefore, the 1-octen-3-ol produced by soil microorganisms, including ECM fungi, by interfering in fungal growth may modulate the structure and composition of ECM fungal communities belowground. ECM fungi produce 1-octen-3-ol mainly on their fruitbodies, an area with a high number of spores, similarly to what occurs in the conidial masses of microfungi [12]. It is known that 1-octen-3-ol can inhibit fungal growth and spore germination of several microfungi [3,11,12,14]. For example, in *Penicillium paneum*, 1-octen-3-ol is a self-inhibitor volatile that blocks the germination process, and, in *Monilinia fructicola,* 1-octen-3-ol destroys the hyphae morphology and cell structure [14]. It is suggested by Chitarra and colleagues [12] that this C-8 volatile has a common function as an inhibitor of premature spore germination. In microfungi, spores and conidial masses are directly exposed to the air, so a volatile produced in the fruitbody or by conidia could be a more efficient self-inhibitor compound of the germination until more appropriate environmental conditions prevail [12]. Moreover, in *Agaricus bisporus*, it is an effective autoregulator of fruiting body development [41].

Our experimental design did not allow us to distinguish between a self-inhibitory behavior or a lower threshold, but clearly showed *B. reticulatus* is more sensitive to 1-octen-3-ol than *L. deliciosus* and *T. leptoderma*. The effects of 1-octen-3-ol on mycorrhizal sporocarps have been studied to a lesser extent compared to other fungal gilds. However, a recent study showed that VOCs can act as an attractant for certain insects, which is important for spore dispersal and spore germination of some mycorrhizal fungi [42]. In addition, 1-octen-3-ol was identified during the *T. borchii*–*Tilia americana* symbiosis process [22], showing that this and other VOCs can have a role on symbiosis communication. Three VOCs (1-pentanol, 2,3-dimethyldecane and *p*-isopropylbenzaldehyde) were found to be involved in pre-symbiotic communication between *Populus* and the ECM fungi *Laccaria bicolor* [43]. Moreover, this ECM fungi produces a sesquiterpene (thujopsene) that increases lateral root formation and root hair length in the pre-symbiotic phase, facilitating mycorrhizal establishment [43].

The fact that we observed seed germination in all treatments for all Cistaceae species showed that the seeds were viable. In general, the 1-octen-3-ol and the ECM fungi resulted in contrasting responses on Cistaceae plant species germination. Although ECM fungi can produce the volatile, the positive effect of the ECM fungi on Cistaceae species germination goes beyond the production of 1-octen-3-ol, as ECM fungal exudates can also promote seed germination [44,45]. Regardless of whether there is mycorrhization or not, the activity and growth of ECM fungi in the soil produces exudates, including VOCs. What our data suggest is that they may have different effects. However, it is also worth noting that more research is needed to understand how fungal exudates and VOCs affect host plants.

On the other hand, the negative effect of 1-octen-3-ol on some Cistaceae species germination may reflect different species-specific sensitivities to the volatile. Inhibitory effects on germination and seedling and vegetative development were reported for *Arabidopsis thaliana* [36,46,47] and *Cistus incanus* [36]. In both plant species, the exposure to low concentrations of 1-octen-3-ol reduced seed germination and, in the seedlings, led to inhibition of root growth and cotyledon bleaching as a result of H_2_O_2_ production [36,46]. Furthermore, in *A. thaliana*, it was observed that even the seeds that were exposed to low 1-octen-3-ol doses were able to germinate when the volatile was removed [36]. This shows that the volatile does not damage the seeds permanently, its presence is required to inhibit germination.

Although the application of the volatile or the ECM fungi had no effect on *C. salvifolius*’ germination, their combined application inhibited seed germination. This contrasting response may reflect the presence of two sources of 1-octen-3-ol (i.e., the added volatile solution and ECM mycelium) that together exceeded this plant species’ threshold. In agreement, inhibitory effects on germination and seedling and vegetative development of *Arabidopsis thaliana* and *Cistus incanus* were observed after exposing the plants to both the synthetic volatiles and to 1 g of truffle [36].

Our data showed that 1-octen-3-ol of external or biogenic origin was able to differently inhibit or stimulate ECM fungi and seed germination, which constitutes evidence for a potential ecological mechanism capable of changing below and aboveground communities (Figure 6). In agreement, *C. incanus* (a truffle host plant of the *Tuber* species) can be inhibited by the volatiles produced by its own symbiont. The truffle hyphae produces 1-octen-3-ol and other volatiles (2-phenylethanol, 3-methyl-1-butanol, 1-hexanol, 3-octanol, 3-octanone and trans-2-octenal) that are considered phytotoxic due to their ability to harm plants [21,36]. They are key to the formation of so-called “burnt areas” in truffle orchards, an area with no, or limited, vegetation around the mycorrhizal plant or tree [8]. These volatile organic compounds were also reported in *Terfezia* species [48]. 

Low concentrations of this volatile can also have positive effects. For example, at a low concentration (0.05µL/L), 1-octen-3-ol showed plant growth-promoting activity in tomato plants by enhancing plant height, basal stem diameter, root number, fresh weight and dry weight [17]. In addition, in *A. thaliana*, 1-octen-3-ol improves resistance of mature plants to *Botrytis cinerea* by activating defence genes [49,50].

As discussed, VOCs, namely 1-octen-3-ol, exert several effects, including growth suppression or induction in both plants and fungi [21,36], induction of defensive behaviors [9,10] and inhibition of spore [3,11,12,14] and seed germination [36]. The results we obtained suggest that these ECM fungi could modulate below and aboveground communities. Figure 6 integrates the results obtained in the first two assays in a conceptual model about the putative effects of 1-octen-3-ol on the ECM fungi and Cistaceae community (below and aboveground communities) in a Mediterranean shrubland. The application of 1-octen-3-ol influences the relative abundance of ECM fungi and of the host (Cistaceae) species in a species-specific way. Further, from our results, we conclude that the interplay between ECM fungi and host plants is much more complex than just a communication based on 1-octen-3-ol. Since the relative abundance of host plants tends to decrease in the presence of the volatile, while it increases in presence of ECM fungi, our results clearly show that 1-octen-3-ol has a role in modulating below and aboveground communities. However, we are still at the beginning of the path leading to the understanding of the complete mechanism involved in this co-regulation.

## 5. Conclusions

As hypothesized, both the ECM fungal species and the Cistaceae species showed different thresholds to the low 1-octen-3-ol dose. Seed germination and plant development of Cistaceae species were influenced by ECM fungal volatiles, such as 1-octen-3-ol. Similar to previous studies, we observed that a low dose of this volatile inhibited or delayed seed germination. This inhibition could be part of the ECM fungal life cycle regulation, namely during fruiting season. During the development of the fruitbody, the fungus produces volatiles, such as 1-octen-3-ol, that lead to the formation of “burnt areas”. In these areas, the germination of host and other plants are inhibited or delayed, which allows the mycelium to spread in the soil and creates space for the frutification of the fungus. Further research is needed to understand the mechanisms underlying this phenomenon and how ECM fungal volatiles influence their plant host development and below and aboveground communities.

## Figures and Tables

**Figure 1 jof-09-00180-f001:**
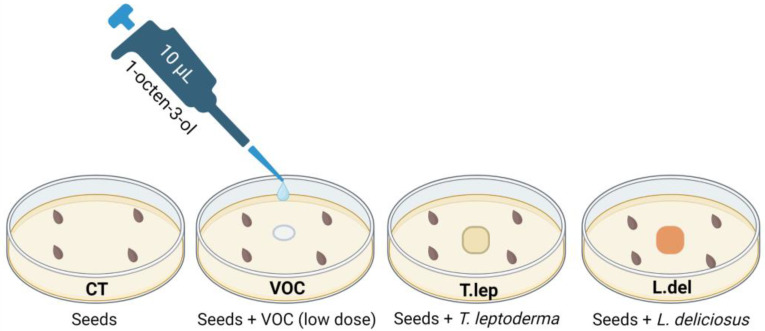
Experimental design to evaluate the effect of 1-octen-3-ol and ECM mycelium on Cistaceae species germination (N = 5).

**Figure 2 jof-09-00180-f002:**
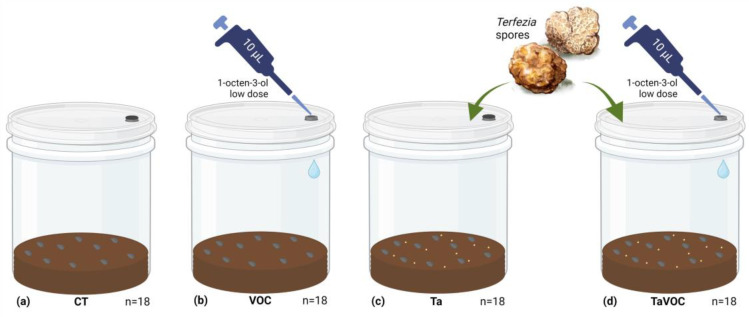
Experimental design to evaluate the effect of 1-octen-3-ol and *Terfezia* spores on *Cistus salviifolius* germination and plant traits. (**a**) Control treatment only with *Cistus salviifolius* seeds; (**b**) treatment with 10 µL of 1-octen-3-ol low dose; (**c**) treatment with 10 mL of *Terfezia* sporal inoculum (10^8^ mL^−1^); (**d**) treatment with 10 µL of 1-octen-3-ol low dose and 10 mL of *Terfezia* sporal inoculum (10^8^ mL^−1^). Ten seeds previously sterilized were distributed in each container (N=18 per treatment).

**Figure 3 jof-09-00180-f003:**
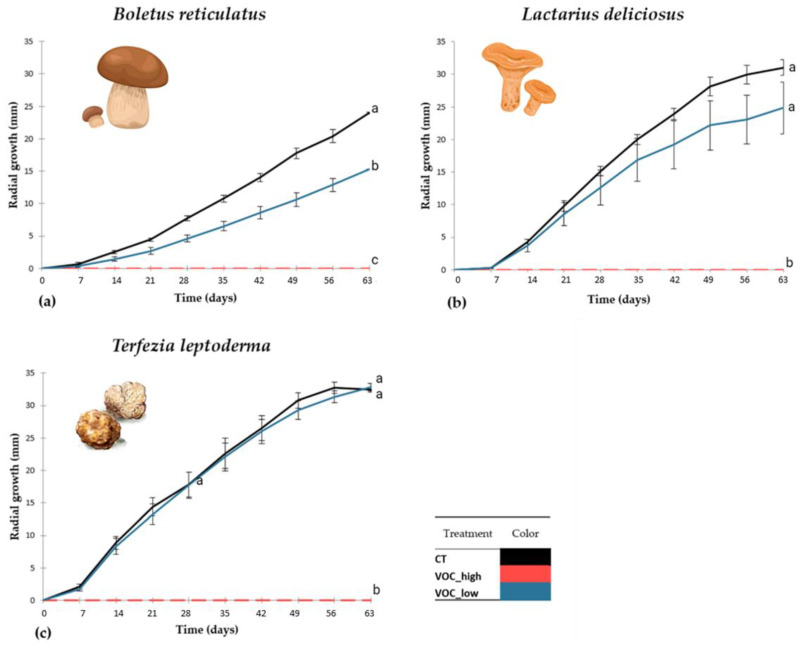
Effects of the 1-octen-3-ol doses on mycelium growth of *Boletus reticulatus*, *Lactarius deliciosus* and *Terfezia leptoderma* (N = 10). (**a**) Mycelium growth of *B. reticulatus*; (**b**) mycelium growth of *L. deliciosus*; (**c**) mycelium growth of *T. leptoderma*, measured over 63 days. Treatments: CT—black line; VOC_high—red line; VOC_low—blue line. Data were compared using a Kruskall–Wallis test. Post hoc comparisons were made using a Dunn’s test. Data are mean ± standard error. Different letters show significant differences (*p* < 0.05) between treatments for each ECM fungus.

**Figure 4 jof-09-00180-f004:**
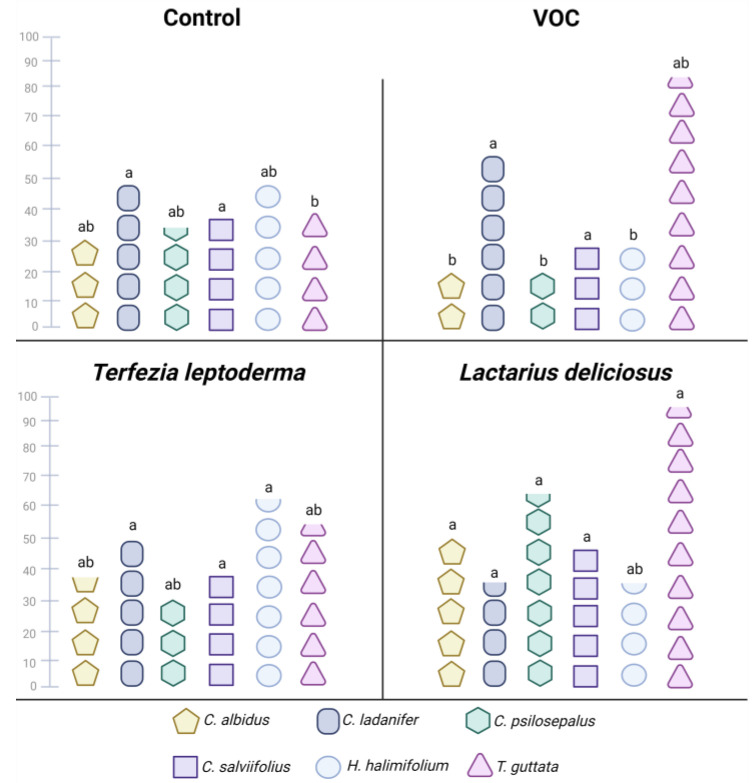
Effects of 1-octen-3-ol 1 µM (VOC) and ECM mycelium (*T. leptoderma* and *L. deliciosus*) on Cistaceae germination rates (%, N = 5). Data were compared using a Kruskall–Wallis test. Post hoc comparisons were made using a Dunn’s test. Data are means. Different letters show significant differences (*p* < 0.05) between treatments for each Cistaceae species.

**Figure 5 jof-09-00180-f005:**
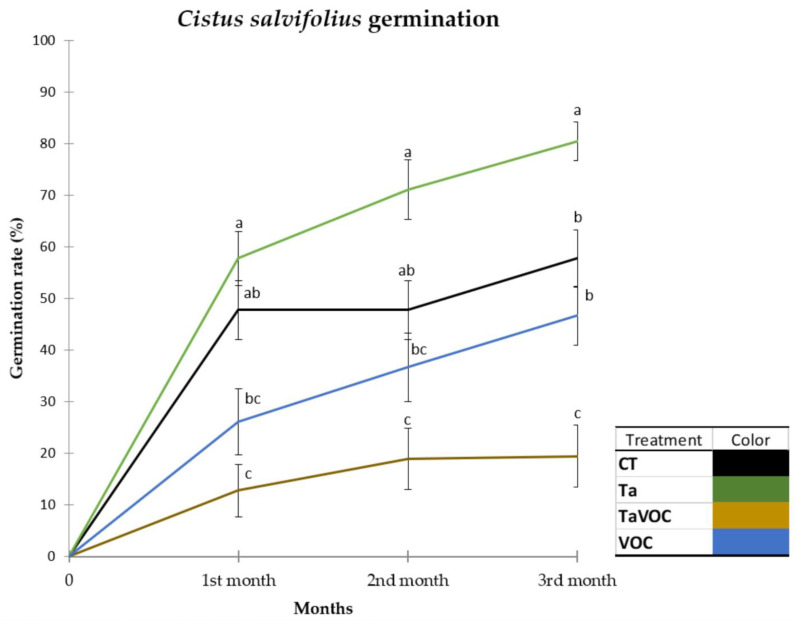
*Cistus salviifolius* seeds germination over three months, showing the effect of *Terfezia arenaria* inoculum and 1-octen-3-ol on *Cistus salviifolius* (N = 18 per treatment). Treatments: CT—control; VOC—1-octen-3-ol 1 µM; Ta—*Terfezia arenaria* inoculum; TaVOC—*T. arenaria* inoculum and 1-octen-3-ol 1 µM. Data were compared using a Kruskall–Wallis test. Post hoc comparisons were made using a Dunn’s test. Data are mean ± standard error. Different letters show significant differences (*p* < 0.05) between treatments.

**Figure 6 jof-09-00180-f006:**
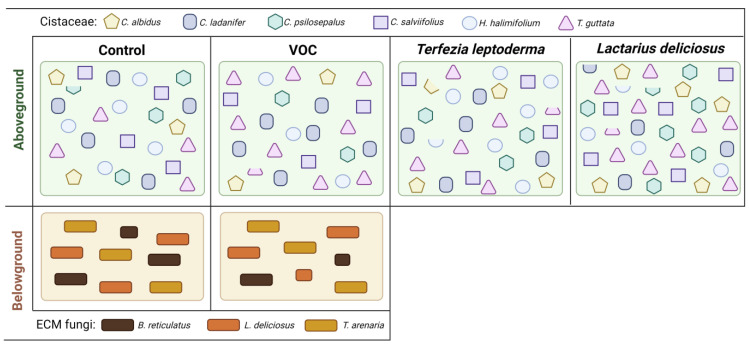
Illustration of the effects of 1-octen-3-ol, of external or biogenic origin, on below and aboveground communities, namely on the ECM mycelial growth and Cistaceae sp. germination.

**Table 1 jof-09-00180-t001:** Effects of *Terfezia arenaria* inoculum and 1-octen-3-ol on *Cistus salviifolius* plant traits: shoot length (SL), shoot weight (SW), root length (RL), root weight (RW), number of lateral shoots and number of leaves observed six and nine months after the inoculation. (N = 6 per treatment). Treatments: CT—control; VOC—1-octen-3-ol 1 µM; Ta—*Terfezia arenaria* inoculum; TaVOC—*T. arenaria* inoculum and 1-octen-3-ol 1 µM.

		6 Months			9 Months		
Trait	Treatment	Mean	SD		Mean	SD	
Shoot length (cm)	CT	4.34	±3.14		8.91	±4.09	
	Ta	5.63	±3.26		10.38	±4.33	
	TaVOC	4.83	±3.04		11.40	±4.69	
	VOC	4.35	±2.61		7.54	±4.29	
Root length (cm)	CT	6.11	±3.31		9.28	±4.13	
	Ta	4.83	±2.25		7.23	±2.88	
	TaVOC	3.81	±1.74		6.94	±2.47	
	VOC	4.75	±3.22		5.94	±2.20	
Shoot fresh weight (mg)	CT	52.11	±46.14		99.75	±74.21	
	Ta	86.35	±84.16		146.93	±144.51	
	TaVOC	149.33	±129.22		309.02	±245.66	
	VOC	53.80	±45.01		89.45	±82.70	
Root fresh weight (mg)	CT	16.84	±14.76		42.35	±40.05	
	Ta	21.65	±20.46		46.16	±43.21	
	TaVOC	32.35	±31.13		58.74	±34.40	
	VOC	12.02	±13.01		44.85	±40.45	
Branches (No/shoot)	CT	0.00	±0.00	b	0.81	±0.61	b
	Ta	0.32	±0.07	b	4.05	±3.52	a
	TaVOC	1.57	±0.61	a	5.00	±4.90	a
	VOC	0.38	±0.09	b	1.05	±0.74	a,b
Leaves (No/shoot)	CT	10.90	±5.58	a	11.39	±5.96	b
	Ta	13.71	±5.05	a	28.10	±19.35	a
	TaVOC	13.55	±5.97	a	49.43	±39.12	a
	VOC	12.24	±5.64	a	13.24	±4.88	a,b

Data were compared using a Kruskall–Wallis test. Post hoc comparisons were made using a Dunn’s test. Data are mean ± standard deviation (SD). Different letters show significant differences (*p* < 0.05) between treatments.

## Data Availability

Not applicable.

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
