# Peer review of "The Potential of Ectomycorrhizal Fungi to Modulate below and Aboveground Communities May Be Mediated by 1-Octen-3-ol"

_jof, 2023, doi:10.3390/jof9020180_

Round 1
Author Response
Authors: The authors would like to thank the reviewers for taking the time to review our manuscript, for their attentive analysis and for the positive comments that helped improve this manuscript. It is a great pleasure to have one's work so thoroughly evaluated and critiqued.
Our responses are presented below the reviewer’s comments.
Comments
The paper provides a good number of explanatory figures which are very useful for understanding.
Please, Check scientific names in italics
Line 32: "carpophores" instead of "mushrooms".
Authors: We agree with the Reviewer’s suggestion. However, we consider that “sporocarps” is the best definition to include both basidiomycetes and ascomycetes, addressed in this work. The word “mushrooms” was changed to “sporocarps”; this change was applied throughout the manuscript.
Line 66: two ECM fungi per treatment
Authors: We agree with the Reviewer and changed accordingly.
Line 69: Why was T. arenaria chosen and not one of the previously selected ECM species for the germination test; there is no logical correlation between the germination test and the growth test. A priori, the effect of one Terfecia species or another on Cistus salviifolius is not the same; the authors should justify this.
Authors: We agree with the Reviewer and therefore a justification was added to the manuscript. The original idea of this work was to use Terfezia arenaria in all experiments. After the fieldwork, we proceeded with the morphological identification of the collected and isolated specimens. When the first experiments were running, due to the similarity between Terfezia species, we made the molecular identification of these samples. Although most of the Terfezia ascocarps harvested were Terfezia arenaria, the ascocarp successfully isolated for pure culture was Terfezia leptoderma. So, after field collection, we found that we had viable sporal inoculum of T. arenaria. However, we did not have pure culture. Since both species, T. arenaria and T. leptoderma are characteristic of acid soils and have the same hosts, the Cistaceae and the experiments were already running, we decided to use both species - Terfezia leptoderma in pure culture and Terfezia arenaria in the sporal inoculum.
Line 86: Specify whether it is per ml, ul or what, throughout the manuscript.
Authors: We agree with the Reviewer. This change was made throughout the manuscript.
Line 111: Were antibiotics added to the PDA nutrient medium? Were there any contaminations?
Authors: We agree with the Reviewer’s comment. No antibiotics were added to the PDA medium. No contaminations were observed due to the use of pure cultures of the ECM fungi, and the VOC solution was sterilized by filtration. This was clarified in the text.
Line 110: Specify whether it is ug per ul, ug per plate or what, throughout the text;
Authors: We agree with the Reviewer. This change was made throughout the manuscript.
the authors should briefly justify these doses, and why they are classified as low or high.
Authors: We agree with the Reviewer and clarified this in the text. We had reviewed studies testing this compound (1-octen-3-ol) and surveyed the concentrations used [4,18–21]. The concentrations tested in these studies were verified with positive results, and a concentration that fell within that reported range was selected — this is the value that we consider to be the low VOC. While the high dose is higher than all the doses reported.
Since we could not measure the amount of volatile released in each plate, and the reported studies referred to the applied concentration and not to the concentration in the petri dish (or other container), we decided to designate the amount as doses instead of concentration. This way, we can know the amount of volatile placed in each plate/container where the test is carried out.
Line 128: Why have no limiting doses for plants been used to test the buffering effect of ECM? Explain
Authors: We understand the Reviewer’s comment. However, in the first experiment, we observed that the high dose inhibited the growth of the three ECM fungi. As we intended to study the interaction between the volatile, ECM fungus and the plant, we decided not to use the high dose so that it would not inhibit the fungus in those two assays.
Line 134: It is not clear whether VOC is supplied at the edge or in the centre of the plate; in the figure it seems to be placed in the centre of the plate; ¿why at the edge and not in the centre? There is no equal distance with the 4 seeds
Authors: We agree with the Reviewer. The VOC in this assay was supplied in the centre of the plate, it was rectified in the manuscript. We thank the referee for clarifying this issue.
Line 135: Adding a non-chloroform control would have improved the experimental design.
Authors: We agree with the Reviewer and will take this suggestion into consideration in future works.
Line 139: What is the relative humidity of the chamber? Will the seeds be able to germinate and the seedlings survive in PDA medium without watering after 35 days? Justify
Authors: We agree with the Reviewer’s comment. The relative humidity of the chamber was 60 ± 5%; this information was added to the manuscript. The PDA medium contains enough water content to support mycelium and/or seed development for at least 2 months. Therefore we consider that plant growth was not affected by humidity.
Line 149: clarify if low dose is 0.17 ug.
Authors: We agree with the Reviewer. It was clarified in the manuscript.
Line 156: Describe "aseptic environment".
Authors: We agree with the Reviewer’s comment. The seeds were distributed in each container inside a flow chamber. This information was added to the manuscript, and “aseptic environment” was eliminated to avoid confusion.
Line 163: The growing medium should always have been in darkness, for a correct growth of the plant and the fungus. Justify
Authors: We understand the Reviewer’s comment. We applied a 16/8 dark/light photoperiod to have more time of darkness for the correct growth of the plant and the fungus. Also, the bottom part of the microboxes, containing the substrate with the seeds and the spores, was covered with foil to decrease light incidence in this area. This information was lacking in the methodology, so it was added to the manuscript.
Line 165: 10 Cistus seedlings growing in a 9 cm diameter box for 9 months; was this enough growing space? Explain.
Authors: We understand the Reviewer’s comment. The microboxes used in this assay had a base of 90 mm diameter, 120 mm of height and 118 mm cover diameter and a total volume of 870 ml. This volume and height allowed the correct development of the Cistus seedlings during this period.
6 microboxes per treatment?
Authors: We agree with the Reviewer. 6 microboxes per treatment was added to the manuscript.
CHECK TABLES AND FIGURES: missing information on N, meaning of SD, meaning of letters, type of test or p-value.
Authors: We agree with the Reviewer. These changes were made throughout the manuscript.
Reviewer 2 Report
The manuscript is about on the effects of 1-octen-3-ol compound in some ECM fungi and both in seed germination of Cistaceae plant species. I have some questions about general reading. First of all, none of the results of the study support the title of the manuscript. The abstract should be more in line with the results obtained.
Indicate in Materials and Methods how the ECM species tested have been identified.
The number of containers per treatment to be used in assay 3 should be indicated in the material and methods section.
In the discussion, it does not make much sense to name non-ectomycorrhizal plant species as an example for the effects of volatile compounds. It would make more sense to find literature on other truffle plant species (black truffle, ...) and mycorrhizal mushrooms.
In addition, two ECM species, Lactarius and Boletus, are tested, which commonly fruit under mycorrhization of other host plant species (pinus, quercus), which have not been analysed.
--------
>Line 17-18: This sentence must be removed: “In general, the presence of the ECM fungi stimulated seed germination while that of 1-octen-3-ol inhibited it.” According to table S1, there are no significant differences to give this conclusion. Only a positive effect on Tuberaria guttata by Lactarius deliciosus is found. No clear conclusions can be drawn from the results of bioassay 2, as there are no significant differences compared to the control (with some exceptions). The number of replicates is low considering that these are in vitro assays where they could easily have at least 15-20 replicates per treatment.
>Line 18-20: This sentence should be modified: “The combined application of the ECM fungus and the volatile further inhibited seed germination possibly due to the accumulation of 1-octen-3-ol above the plant species’ threshold.” In other words, it seems that the inoculum of T. arenaria is producing this compound. Is there any evidence for this? In this time (0-3 months) was spore germination controlled, was there mycelium, was there root mycorrhization? No data is given for this information. According to Figure 5, it seems that VOC inhibits the development of T. arenaria. Why was this species not used in bioassay 1? That would give information about what is happening during bioassay 3.
>Line 86: 1x108 spores?
>Line 110-111: why those doses? Any reference to this?
>Line 163: 16/8 dark/light or 16/8 light/dark ?
Author Response
The authors would like to thank the Reviewer for the positive feedback and the suggestions made, which have significantly improved this work. We have commented below on each of the points raised by the Reviewer.
The manuscript is about on the effects of 1-octen-3-ol compound in some ECM fungi and both in seed germination of Cistaceae plant species. I have some questions about general reading. First of all, none of the results of the study support the title of the manuscript. The abstract should be more in line with the results obtained.
Authors: We understand the Reviewer’s comment. However, we consider it is a consequence of a lack of discussion about the results from which outcomes the last figure concerning the effects of 1-octen-3-ol on below- and aboveground communities. During the revision process, we developed the discussion of this topic. We understand the comment about the abstract, which was revised accordingly with the reviewer’s suggestions and the limit of words.
Indicate in Materials and Methods how the ECM species tested have been identified.
Authors: We agree with the Reviewer. The information was added in the Fungal material chapter.
The number of containers per treatment to be used in assay 3 should be indicated in the material and methods section.
Authors: We agree with the Reviewer. This information was added to the material and methods section.
In the discussion, it does not make much sense to name non-ectomycorrhizal plant species as an example for the effects of volatile compounds. It would make more sense to find literature on other truffle plant species (black truffle, ...) and mycorrhizal mushrooms.
Authors: We agree with the Reviewer. But unfortunately, the effects of 1-octen-3-ol on mycorrhizal plants have been studied to a lesser extent compared to its effects on non-mycorrhizal plants and AMF. However, some research on Cistus incanus (a plant host of Tuber melanosporum) is referred to in the manuscript. In the same study with C. incanus was also used th A. thaliana (non-ectomycorrhizal plant) to evaluate the effects of 1-octen-3-ol. We also, consider it important to mention literature results on this plant due to its importance as a model plant, so often used in similar studies.
The effects of 1-octen-3-ol on mycorrhizal mushrooms have also not been well studied. However, some research has suggested that 1-octen-3-ol may positively affect mycorrhizal mushrooms. For example, it has been found that 1-octen-3-ol can act as an attractant for certain insects, which is important for spores’ dispersal and spore germination of some mycorrhizal fungi.
In the manuscript was added this and other relevant information as suggested by the reviewer.
In addition, two ECM species, Lactarius and Boletus, are tested, which commonly fruit under mycorrhization of other host plant species (pinus, quercus), which have not been analysed.
Authors: We agree with the Reviewer. However, in the case of Cistus plants, they have been found to form mycorrhizal relationships with various species of fungi, including those in the genus Terfezia, Lactarius and Boletus (Comandini et al. 2006; Martins et al 2019; Águeda et al. 2017). Therefore, we selected Cistaceae species because they are more straightforward to handle under in vitro conditions. Further, this study is part of a doctoral thesis, which aims to enhance the knowledge of the ECM fungal community in Mediterranean shrublands, where the Cistaceae are more abundant in the plant community than Pinus or Quercus. A paragraph was added to the introduction, contextualizing Cistus and the ECM fungi.
--------
>Line 17-18: This sentence must be removed: “In general, the presence of the ECM fungi stimulated seed germination while that of 1-octen-3-ol inhibited it.” According to table S1, there are no significant differences to give this conclusion. Only a positive effect on Tuberaria guttata by Lactarius deliciosus is found. No clear conclusions can be drawn from the results of bioassay 2, as there are no significant differences compared to the control (with some exceptions). The number of replicates is low considering that these are in vitro assays where they could easily have at least 15-20 replicates per treatment.
Authors: We agree with the Reviewer. If we had more replicates per treatment, we would have a clear answer for each treatment. Unfortunately, it was not possible due to some constraints. However, we also understand that we cannot take that conclusion, so we revised the sentence on the abstract to be supported by the results.
>Line 18-20: This sentence should be modified: “The combined application of the ECM fungus and the volatile further inhibited seed germination possibly due to the accumulation of 1-octen-3-ol above the plant species’ threshold.” In other words, it seems that the inoculum of T. arenaria is producing this compound. Is there any evidence for this?
Authors: We understand the Reviewer’s point. However, we did not measure the amount of 1-octen-3-ol produced by T. arenaria mycelium. In another study, we identified 1octen-3-ol released by T. arenaria ascocarps, representing more than 60% of the total volatiles identified. The volatile was also identified in other Terfezia species (Kamle et al. 2017; Farag et al. 2021) and is common in mushrooms. In the case of mycelium, there have been no studies published proving the production by Terfezia. However, is known that 1-octen-3-ol can be produced by the mycelium of several mushroom species (Belinky et al. 1994; Schindler and Seipenbusch 2009), and in particular in Tuber borchii (Bianchetto truffle)(Splivallo et al. 2007). Also, this compound was identified during T. borchii–Tilia americana symbiosis process (Menotta et al. 2004). Taking this into account, we believe that this, volatile is (among others) produced by the T. arenaria mycelium.
This information was integrated into the manuscript.
In this time (0-3 months) was spore germination controlled, was there mycelium, was there root mycorrhization? No data is given for this information.
Authors: We agree with the Reviewer. From the analysis of the root samples in the third month, for the treatments with T. arenaria inoculum, mycorrhizal structures were detected in the initial stage. This information was added to the manuscript.
According to Figure 5, it seems that VOC inhibits the development of T. arenaria. Why was this species not used in bioassay 1? That would give information about what is happening during bioassay 3.
Authors: We agree with the Reviewer. Therefore, a justification was added to the manuscript.
The original idea of this work was to use Terfezia arenaria in all experiments. After the fieldwork, we proceeded with the morphological identification of the collected and isolated specimens. Then when the first experiments were running, due to the similarity between Terfezia species, we made the molecular identification of these samples. Although most of the Terfezia ascocarps harvested were Terfezia arenaria, the ascocarp successfully isolated for pure culture was Terfezia leptoderma. So, after field collection, we found that we had viable sporal inoculum of T. arenaria. However, we did not have pure culture.
Both species, T. arenaria and T. leptoderma are characteristic of acid soils and have the same hosts, the Cistaceae. Thus, due to the similarity, we chose to use both species - Terfezia leptoderma in pure culture and Terfezia arenaria in the sporal inoculum.
>Line 86: 1x108 spores?
Authors: We agree with the Reviewer. 1x108 spores per ml. This information was added to the manuscript.
>Line 110-111: why those doses? Any reference to this?
Authors: We agree with the Reviewer. The authors reviewed studies testing this compound (1-octen-3-ol) and surveyed the concentrations used [4,18–21]. The concentrations tested in these studies were verified with positive results, and a concentration that fell within that reported range was selected —this value we consider to be the low VOC. While the high dose is higher than all the doses reported.
Since we couldn't measure the amount of volatile released in each plate, we chose to designate the amount as doses instead of concentration. This way, we can know the amount of volatile placed in each plate/container where the test is carried out.
A brief justification of the doses was inserted in the manuscript.
>Line 163: 16/8 dark/light or 16/8 light/dark ?
Authors: We agree with the Reviewer. It was applied a 16/8 dark/light photoperiod in order to have more time of darkness, for the correct growth of the plant and the fungus. Also, the bottom part of the microboxes, containing the substrate with the seeds and the spores, was covered with foil to decrease light incidence in this area. This information was lacking in the methodology, so it was added to the manuscript.
Reviewer 3 Report
The manuscript presented that three in vitro assays to (i) explore the effects of 1-octen-3-ol on mycelium growth of three ECM species: Terfezia leptoderma, Lactarius deliciosus and Boletus reticulatus; (ii) investigate the impact of the 1-octen-3-ol and the three ECM species on the germination of six host Cistaceae species; (iii) study the impact of the 1-octen-3-ol or ECM fungus (Terfezia arenaria) on host plant Cistus salviifolius traits along nine months. This work provide very interesting data.
But in its current state, there are some issues worth thinking about and discussing. I disagree with the statement that 1-octen-3-ol may mediate changes in below and aboveground communities mentioned many times in this article. In my view, the author only simply analyzed the effects of 1-octen-3-ol and/or ECM fungi on different fungi or plant traits, which cannot reflect the characteristics of below and aboveground communities. In the third experiment, why change the species of ECM fungus, this needs an explanation…
A couple of specific and minor points:
The Abstract section should provide the background of this study in a sentence or two and simplify the research objectives. The focus should be on findings and implications. In addition, the VOC that appears for the first time in the abstract should be described as the full name.
Line 44. “Botrytis cinerea” should be italicized.
Line 45-46. However, while low concentrations of 1-octen-3-ol activate plant defense, at high concentrations can damage plants. There are redundant words…I guess this sentence needs to be checked.
Line 11 and 64. “in vitro” should be italicized.
In the 2.3.1. Mycelium development of Materials and Methods section: three doses were chosen, 0 µg (designated as CT), 0.17 µg (designated as VOC_low) and 1280 µg (designated as VOC_high). Can you explain the rationale for dose selection? Have you ever determined how much 1-octen-3-ol ECM fungi can release?
In your second experiment, i.e, 2.3.2. Effect of VOC and ECM fungi on Cistaceae species germination, why no treatment for the effect of combination of VOC and ECM on Cistaceae species germination? This deserves a comment.
Figure 5 and Table 1 should add a note of significance. Other data in Table 1, such as Shoot length, Root length… I guess need to add significance.
Line 234. The citation for tables in the text need to be carefully checked. I guess the sentence should refer to Table 1. In the table legend you provided in the Supplementary Materials section, there is no table S1, but table S1 was quoted in the text.
Line 243-244. From the Table 1 provided in this paper, it is known that the length of shoot and root did not show higher values in the TaVOC treatment. I guess this sentence needs to be revised.
Line 251-253. The fact that the lower dose of 1-octen-3-ol inhibited the mycelium growth of B. reticulatus but not of L. deliciosus and T. leptoderma may reflect a self-inhibitory behavior and/or a lower threshold to the volatile. I think this sentence needs to be split.
Line 268-271. A detailed explanation of the effects of ECM fungal exudates on Cistaceae plant species germination is needed to distinguish the effects of ECM fungi produced the volatile compounds on Cistaceae plant species germination.
Only twenty-five references are cited in this paper, which is not enough for a research article.
Author Response
The authors would like to thank the Reviewer for the positive feedback and the suggestions made, which have significantly improved this work. We have commented below on each of the points raised by the Reviewer.
The manuscript presented that three in vitro assays to (i) explore the effects of 1-octen-3-ol on mycelium growth of three ECM species: Terfezia leptoderma, Lactarius deliciosus and Boletus reticulatus; (ii) investigate the impact of the 1-octen-3-ol and the three ECM species on the germination of six host Cistaceae species; (iii) study the impact of the 1-octen-3-ol or ECM fungus (Terfezia arenaria) on host plant Cistus salviifolius traits along nine months. This work provide very interesting data.
But in its current state, there are some issues worth thinking about and discussing. I disagree with the statement that 1-octen-3-ol may mediate changes in below and aboveground communities mentioned many times in this article. In my view, the author only simply analyzed the effects of 1-octen-3-ol and/or ECM fungi on different fungi or plant traits, which cannot reflect the characteristics of below and aboveground communities.
Authors: We agree with the Reviewer. Therefore, during the revision of the manuscript, we discussed this, to clarify the results obtained and how we interpret them.
In the third experiment, why change the species of ECM fungus, this needs an explanation…
Authors: We agree with the Reviewer. A justification was added to the manuscript.
The original idea of this work was to use Terfezia arenaria in all experiments. After the fieldwork, we proceeded with the morphological identification of the collected and isolated specimens. Then when the first experiments were running, due to the similarity between Terfezia species, we made the molecular identification of these samples. Although most of the Terfezia ascocarps harvested were Terfezia arenaria, the ascocarp successfully isolated for pure culture was Terfezia leptoderma. So, after field collection, we found that we had viable sporal inoculum of T. arenaria. However, we did not have pure culture.
Both species, T. arenaria and T. leptoderma are characteristic of acid soils and have the same hosts, the Cistaceae. Thus, due to the similarity, we chose to use both species - Terfezia leptoderma in pure culture and Terfezia arenaria in the sporal inoculum.
A couple of specific and minor points:
The Abstract section should provide the background of this study in a sentence or two and simplify the research objectives. The focus should be on findings and implications.
Authors: We agree with the Reviewer. The abstract was rewritten.
In addition, the VOC that appears for the first time in the abstract should be described as the full name.
Authors: We agree with the Reviewer. This information was added to the abstract.
Line 44. “Botrytis cinerea” should be italicized.
Authors: We agree with the Reviewer. This change has been made.
Line 45-46. However, while low concentrations of 1-octen-3-ol activate plant defense, at high concentrations can damage plants. There are redundant words…I guess this sentence needs to be checked.
Authors: We agree with the Reviewer. This sentence was removed from the manuscript.
Line 11 and 64. “in vitro” should be italicized.
Authors: We agree with the Reviewer. This change has been made.
In the 2.3.1. Mycelium development of Materials and Methods section: three doses were chosen, 0 µg (designated as CT), 0.17 µg (designated as VOC_low) and 1280 µg (designated as VOC_high). Can you explain the rationale for dose selection?
Authors: We agree with the Reviewer. We reviewed studies testing this compound (1-octen-3-ol) and surveyed the concentrations used [4,18–21]. The concentrations tested in these studies were verified with positive results, and a concentration that fell within that reported range was selected —this value we consider to be the low VOC. While the high dose is higher than all the doses reported.
Since we couldn't measure the amount of volatile released in each plate, we chose to designate the amount as doses instead of concentration. This way, we can know the amount of volatile placed in each plate/container where the test is carried out.
A brief justification of the doses was inserted in the manuscript.
Have you ever determined how much 1-octen-3-ol ECM fungi can release?
Authors: We understand the Reviewer’s point. However, we could not quantify the 1-octen-3-ol ECM released by the fungi during this study. Nevertheless, we consider that it is crucial to fully understand the mechanisms and how the dose/quantity affects plant and fungi development and therefore we are currently developing a protocol to perform this assessment for future works.
In your second experiment, i.e, 2.3.2. Effect of VOC and ECM fungi on Cistaceae species germination, why no treatment for the effect of combination of VOC and ECM on Cistaceae species germination? This deserves a comment.
Authors: We agree with the Reviewer. The authors understand this treatment's issue and pertinence, as this treatment (VOC + ECM) was performed in the third experience. However, in this experiment, we aimed to understand the effect of VOC and ECM alone on seed germination and to compare these two treatments.
We know that fungal mycelium produces volatiles, including 1-octen-3-ol. Therefore, by adding 1-octen-3-ol to the ECM, we could be increasing the concentration of this volatile and would not answer the question of whether or not the effects of the volatile coming from the fungus are similar to the isolated volatile (and at the indicated dose).
With the results obtained, we understand that it will be fundamental in the future to quantify the volatiles produced by the ECM mycelium, namely 1-octen-3-ol. This quantification would allow us to know whether the applied dose will be higher or lower than that produced by the fungus.
Figure 5 and Table 1 should add a note of significance. Other data in Table 1, such as Shoot length, Root length… I guess need to add significance.
Authors: We agree with the Reviewer. This information was added to Figure 5 and Table 1.
Line 234. The citation for tables in the text need to be carefully checked. I guess the sentence should refer to Table 1. In the table legend you provided in the Supplementary Materials section, there is no table S1, but table S1 was quoted in the text.
Authors: This sentence refers to Table S1, provided in the Supplementary Materials section. The legend of Table S1 in the Supplementary Materials section was updated.
Line 243-244. From the Table 1 provided in this paper, it is known that the length of shoot and root did not show higher values in the TaVOC treatment. I guess this sentence needs to be revised.
Authors: We agree with the Reviewer. The sentence was revised accordingly.
Line 251-253. The fact that the lower dose of 1-octen-3-ol inhibited the mycelium growth of B. reticulatus but not of L. deliciosus and T. leptoderma may reflect a self-inhibitory behavior and/or a lower threshold to the volatile. I think this sentence needs to be split.
Authors: We agree with the Reviewer. This sentence was rewritten. “The lower dose of 1-octen-3-ol inhibited the mycelium growth of B. reticulatus but not of L. delicious and T. leptoderma. This may reflect self-inhibitory behaviour and a lower threshold to the volatile.”
Line 268-271. A detailed explanation of the effects of ECM fungal exudates on Cistaceae plant species germination is needed to distinguish the effects of ECM fungi produced the volatile compounds on Cistaceae plant species germination.
Authors: We agree with the Reviewer. Therefore, we developed this point in the discussion.
Only twenty-five references are cited in this paper, which is not enough for a research article.
Authors: We agree with the Reviewer. Therefore, during the manuscript’s revision, more references were added with pertinent information to the manuscript.
Round 2
Reviewer 2 Report
In general, I am grateful for the authors' responses, as well as for the information added to the manuscript. I am aware of the difficulty of pure cultivation of this type of mushroom species. Moreover, this study highlights some aspects that could be developed and further evaluated in the framework of a doctoral thesis.
Reviewer 3 Report
The manuscript has revised according to the previous suggestions and comments and could be published.